# Estimation of the Achievable Performance of Mobile Ad Hoc Networks with Optimal Link State Routing

**Gennady Kazakov**

Moscow Aviation Institute (National Research University), Volokolamskoe Shosse 4, 125993 Moscow, Russia;
jee2@mail.ru

**Abstract:** The paper explores the challenges of constructing self-organizing wireless mobile ad hoc networks (MANETs) utilizing Optimal Link State Routing (OLSR) with MPR (MultiPoint Relay) optimization and quality control through the RSVP (Resource Reservation Protocol). Analytical expressions are presented for calculating the achievable network characteristics, including route acquisition time, network efficiency (routing overhead), packet transmission delay (end-to-end delay), and signal propagation losses between nodes assuming no packet collisions within the network nodes. The possibility of network scalability is analyzed depending on the scenarios of operation and the number of network nodes. Recommendations for the construction and scalability of self-organizing wireless networks are formulated based on the conducted evaluations and calculations.

**Keywords:** MANET; IEEE 802.11; OLSR protocol; RSVP protocol; network scenario; network efficiency characteristics; scalability

## 1. Introduction

There are many situations where users cannot rely on telecommunication infrastructure. Infrastructure may be too costly to deploy or may be completely absent. In such cases, mobile ad hoc networks (MANETs) can provide a solution for establishing communication between devices in the absence of base stations or access points. Devices in these mobile networks can send data from one device to another remote device through several other devices. Some devices can function as end transmitters or end receivers, while others can act as relays, and some can operate as both relays and receivers/transmitters. Some nodes may be connected through wired connections or other high-speed links and can be used as wireless gateways to connect to other networks. A self-organizing communication network is a network with a flexible decentralized infrastructure that changes and distributes functions among nodes when new devices are connected or when there are changes in traffic characteristics, etc. [1].

Unlike infrastructure-based wireless networks, there is no single base station or access point in ad hoc networks that controls the flow of network traffic. Nodes can act as end transmitters, end receivers, and relay functions, creating a distributed wireless network.

Nodes within each other's reach and within the range of a neighboring node can exchange packets without the assistance of any other entity. To transmit packets to nodes further away, nodes that are situated between the source and destination nodes relay packets from one to another, similar to traditional routers, to reach the final destination. The evaluation of the efficiency of construction and operation of self-organizing wireless networks is quite relevant due to the constantly expanding areas of application for an MANET:

1. Emergency services: the network can be used in emergency or rescue operations as a replacement for stationary infrastructure or in areas where no infrastructure is available, to provide assistance, for example, in firefighting, flooding, earthquakes, etc.

2.  Coverage extension: expanding access to cellular networks, internet connectivity, external networks, etc.
3.  Sensor networks: indoor smart sensors and actuators embedded in consumer electronics; monitoring environmental data and animal movements; chemical or biological detection.
4.  Specialized networks for meetings or lectures: peer-to-peer wireless networks; using wireless networks in homes or offices; conferences, exhibitions, presentations, etc.
5.  Context-aware services: additional services such as call forwarding and a mobile workspace; location-based services; time-dependent services.
6.  Information and entertainment sector: tourist information, amusement parks, multiplayer games, sports stadiums, trade fairs, shopping malls, etc.; networks within airports.
7.  Commercial and civil sectors: e-commerce—electronic payments anytime and anywhere; business—dynamic access to databases and mobile offices; transportation—road or emergency assistance and transmission of road and weather conditions, taxi networks, and intercity transportation.

The first mobile peer-to-peer networks were initially developed for voice communication using narrowband channels in the context of automated military control systems—Mobile Packet Radio Networks [1,2]. These networks allowed for packet data transmission in-between voice transmission using ALOHA [3] or CSMA (Carrier Sense Multiple Access) [4] mechanisms for channel access. Later, the MIL-STD-188-220 standard [5] was proposed, which describes a protocol stack for organizing mobile peer-to-peer networks using narrowband channels. Additionally, standards such as APCO P25 [6] and TETRA [7] were developed for digital professional radio communication, which, besides voice communication, also supports packet data transmission. The use of narrowband channels in such networks provides a sufficiently long-range reception, eliminating the need for intermediate relays. However, these networks have low throughput capacity (up to 10 kbps).

Currently, due to the rapid growth in the areas of application for an MANET, the number of transmitted traffic types is increasing. This necessitates the need to increase the network's throughput, i.e., the use of broadband communication channels. However, this leads to a decrease in the transmission range between nodes, requiring nodes to also act as relays to expand the coverage area of the peer-to-peer network. This makes the network multihop and necessitates the use of specialized routing protocols.

The development of broadband mobile peer-to-peer networks has also actively begun for military telecommunications applications. In particular, within the European Secure SOftware defined Radio (ESSOR) program, the High Data Rate Wave Form (HDRWF) standard [8] was developed, which describes the protocol stack for organizing broadband mobile peer-to-peer networks. A similar standard, Wideband Networking Waveform (WNW), was developed as part of the Joint Tactical Radio System (JTRS) program [9] by the US Armed Forces. The WNW standard utilizes the Dynamic Time Division Multiple Access (DTDMA) mechanism [10] for deterministic channel access and the proactive Radio Open Shortest Path First (ROSPF) [11] routing protocol.

Civil broadband mobile peer-to-peer networks have gained wide popularity with the introduction of the IEEE 802.11 (Wi-Fi) [12] standard for local wireless networks. In addition to a hotspot mode with an access point, this standard provides two peer-to-peer network modes: ad hoc and mesh. The ad hoc mode allows for organizing only a one-step peer-to-peer network. To extend the coverage area of the ad hoc network, nodes can retransmit packets at the network (IP) layer using dynamic routing protocols. Channel access in the "ad hoc" mode is implemented using the Carrier Sense Multiple Access with Collision Avoidance (CSMA/CA) mechanism.

The Mobile Ad-hoc NETworks Working Group of the Internet Engineering Task Force (IETF) [1] is responsible for addressing routing protocol development and network evolution issues. Recommendation [1] analyzes the routing issues in MANETs and proposes

possible qualitative and quantitative metrics for evaluating the performance and efficiency of these protocols.

Routing protocols for mobile self-organizing networks can be classified into four main groups: protocols with proactive routing, protocols with reactive routing, hybrid protocols, and protocols that use geographical position data of nodes. Among the most commonly used protocols are proactive protocols: OLSR (Optimized Link State Routing) [13] and B.A.T.M.A.N. (Better Approach To Mobile Ad hoc Networks) [14], and reactive: AODV (Dynamic Source Routing) [15] and LQSR (Link Quality Source Routing) [16]. The selection of the best route between network nodes is based on quality of service (QoS) metrics. These metrics can be measured at the physical, link, and network layers of the OSI model. Numerous studies comparing the listed protocols for different types of networks [17–19] have shown that in MANETs with a limited mobility of nodes, a good enough performance in terms of the speed and quality of transmission, scalability, and other parameters provides protocol OLSR. Therefore, in this study, the OLSR protocol with MPR optimization was chosen for routing. In addition, some decision-making techniques, e.g., the equivalent exchange method [20,21], can be utilized for seeking the compromise while different goals are involved.

A performance analysis and comparison of protocols and networks in general is a challenging task due to the fact that the data transmission process in self-organizing networks is influenced by a large number of different factors, many of which are random in nature. Therefore, due to the dynamic nature and variability of the processes observed in mobile networks, researchers, along with analytical methods of analysis, most often use methods based on network modeling, network emulation, or testbed experiments [22].

At the moment, the market of software packages for network simulation modeling is filled with many solutions. The most popular simulation environments—Simulators of Wireless MANETs—include network simulators Network Simulator 2 (NS-2) and Network Simulator 3 (NS-3) [23], the simulation system GloMoSim (Global Mobile Information System Simulator) [24], the system of the simulation modeling and analysis of communication networks (OPNET Modeler) [25], and a number of others. A comparison of commonly used simulators in an MANET is given in [26]. These tools allow you to specify parameters and technologies at the physical, link, network, and transport layers, as well as the type of traffic at the application layer; select specific node mobility models; and specify various network topology configurations and basic node usage scenarios, including those which stipulate the interoperability of the whole network [27]. Simulation models by their nature cannot provide an accurate description of the network, especially the node mobility behavior and the characteristics of the wireless environment [22].

Network emulation is performed in real-time and brings the network model closer to the real system by combining actual elements of the deployed network implementation with other modelled components [22]. Testbeds use a full set of real components, so the methods of testing network scenarios on a testbed provide the most realistic research results. Recently, when building MANET testbeds, it was suggested to use a universal experimental platform [22,28]. Usually, such a platform is based on a modular architecture, has sufficient versatility, and can be flexibly used to investigate different scenarios of experiments.

One of the actively developing areas in the MANET field is also the development of new routing algorithms based on the predicted location of nodes [22,29]. Such algorithms are expected to be used in specialized networks mainly with rather fast node movement, such as the Vehicle Ad hoc Networks (VANETs) or the special case where vehicles are unmanned devices, the Unmanned Vehicle Networks (UxVs) [22]. In spite of the development of new routing protocols, many research works aimed at improving OLSR continue to appear. For example, an improved OCI-OLSR protocol (Optimized-Control-Interval–Optimized-Link-State-Routing) is proposed in [30], which focuses on better control interval management, enhanced MPR process selection, reducing neighbor hold time, and decreasing flooding. In [31], the multipath heterogenous ad hoc network OLSR (MHAR-OLSR) protocol is described and investigated. MHAR-OLSR is an OLSR extension with new

functionalities: nodes' identification, paths' calculation, paths' classification, and paths' choice, designed for heterogeneous ad hoc networks composed of MANET, VANET, and FANET devices.

Once again, we note that for networks in which the change in node position is quite slow (no faster than the average packet transmission time from source to destination) in most cases, the OLSR protocol or its modifications are used. Thus, in [30], the number of research papers with the keywords "OLSR" and "MANET" is given. This paper studies the research trends of the OLSR routing protocol in MANETs and analyzes numerous advantages of OLSR (low packet delay, ability to work in high-density networks, much higher performance compared to other protocols, etc.). It has been shown that in the time period from 2002 to 2022, the number of research papers on the use of OLSR in MANETs is steadily increasing [30]. Therefore, in this paper, OLSR is selected as the routing protocol for the study.

In practice, there is often a need to quickly evaluate, without simulation, the potential capacity of MANETs to carry traffic corresponding to different classes of service without the detailed definition of all network parameters. In particular, it is of interest to determine the number of nodes or the possible coverage area that can provide, for example, an acceptable packet delay. This problem can be solved by evaluating the "internal" network efficiency metrics proposed in [1,26]. This paper presents analytical expressions for calculating the maximum achievable values of quality metrics for different scenarios of packet transmission in the network using selected routing and Resource Reservation Protocols, assuming no packet collisions in nodes. The obtained estimates allow us to formulate upper limits and constraints on the number of nodes and network scalability for different scenarios of both indoor and outdoor node deployment. The results of the study can form the basis of a methodology to enable the "conscious" or rational selection of MANET parameters for further simulation, network emulation, or testbed experiments under specific operational scenarios or environmental conditions.

The remainder of the paper is organized as follows. Section 2 is a description of the structure of the studied network as well as the OLSR protocol and Resource Reservation Protocol (RSVP). Section 3 discusses the network parameters, various metrics used to analyze the network performance, analytical expressions for calculating the metrics, and results of the calculations. Also, preliminary conclusions and recommendations were formulated based on the results of the calculations. Section 4 analyzes the impact of the calculated parameters on the scalability of the network. Finally, conclusions and suggestions for future work are presented in Section 5.

Table 1 summarizes the abbreviations of terminologies used in this paper.

**Table 1.** The list of abbreviations.

| Notation | Meaning |
|---|---|
| MANETs | Mobile ad hoc networks |
| OLSR | Optimized Link State Routing protocol |
| LSR | Link State Protocol |
| MPR | MultiPoint Relays |
| TC | Topology control |
| QoS | Quality of service |
| RSVP | Resource Reservation Protocol |
| MCS | Modulation and coding scheme |
| GI | Guard Interval |
| IPTD | IP Packet Transfer Delay |
| THPT | Average throughput |
| ETED | End-to-end delay |
| PD | Wi-Fi Protected Access |
| WPA2 | Packet delay |

**Table 1.** *Cont.*

| Notation | Meaning |
|----------|---------|
| CCMP | Counter Mode with Cipher Block Chaining Message Authentication Code Protocol |
| UDP | User Datagram Protocol |
| LOS | Line of sight |
| NLOS | Non-line-of-sight |
| L | Loss |
| dB | Decibel |

## 2. Models and Restrictions

### 2.1. Network Features and Network Structure Description

It is assumed that the network is formed in urban or rural environments only with mobile user terminals (smartphones and laptops) using Wi-Fi technology, in conditions of absence or impossibility to provide communications through the infrastructure of other wireless communication networks—in emergency incidents, in regions with a lack of developed communications infrastructure, as well as in the case of failure of the main communication network.

- The network infrastructure includes only user devices and the communication channels between them.
- The network topology can change, but these changes do not occur faster than the average packet transmission time from source to destination.
- Network devices can be sources of transmitted information or receivers of it, or perform routing functions.
- At the physical layer, the devices are interconnected based on 802.11n and 802.11ac, at center frequencies of 2.4 GHz and 5 GHz, respectively.
- Nodes in the network are assumed to be located both outside and inside buildings.
- The number of user devices in the network is limited by the requirements of a specified quality of the service level.
- The distance between network nodes is chosen based on the permissible level of signal attenuation.
- The routing algorithms used in the network are OLSR with MPR optimization.
- The traffic to be transmitted is VoIP, dialog data, streaming video, and latency-insensitive data.

Figure 1 depicts a diagram of a network segment, including communication nodes and connections between them.

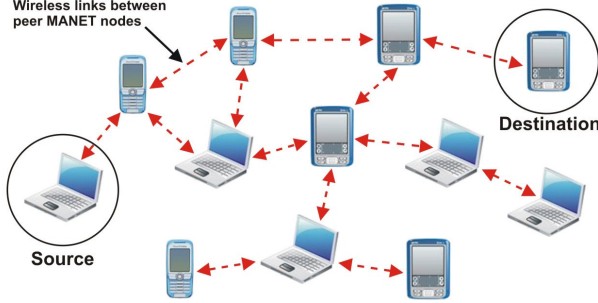

**Figure 1.** Network section diagram.

### 2.2. OLSR Routing Protocol

One of the most effective and commonly used proactive routing protocols for dynamically organized networks with high node density and low node mobility is the Optimized Link State Routing (OLSR) protocol. OLSR performs well in large and dense mobile networks. The protocol is documented in IETF RFC 3626 [13] and RFC 7181 [32]. OLSR is

managed with tables and utilizes an optimization called MultiPoint Relay (MPR) for traffic control. The larger and denser the network, the greater optimization that can be achieved compared to the classical Link State Routing (LSR) algorithm.

OLSR is based on the collection and dissemination of service information about the state of the network. As a result of processing this information, each node can build a model of the current network state in the form of a formal description of the graph, the vertices of which correspond to the network nodes, the edges (or arcs) to the communication lines (links). Having such a graph, any node can calculate the "lengths" of the shortest paths to all addressees in the network and choose the "optimal" route leading to any particular node in the network.

This algorithm reacts well to many unforeseen events, which, first of all, should include spontaneous failures and the recovery of nodes and lines; aggressive effects of the "external environment", leading to the blocking of individual elements of the system; and connections and disconnections of nodes and lines during the operational redeployment of subscribers.

The OLSR protocol employs step-by-step routing, meaning that each node uses its own information for packet routing. Therefore, it is suitable for networks with random and sporadic traffic [33] among a large set of nodes rather than deterministic traffic between a small specific set of nodes. OLSR is also well suited for scenarios where communicating nodes change over time, as no additional control traffic is generated since routes are maintained for all known destinations at all times. Additionally, the protocol provides the advantage of immediate route availability when needed.

To obtain information about the network topology, the OLSR protocol utilizes Hello and control message exchanges. Nodes use this information to determine the next hop in the path of the routed data packet. The OLSR protocol is based on the concept of MultiPoint Relay (MPR). Each node in the network selects several nodes from its neighbors (i.e., nodes with which it has a connection). Consequently, a set of MPR nodes is formed in the network. This set is created in such a way that all nodes within a radius of two hops from a given node (neighbors of neighbors) have symmetric links with the MPR nodes. This means that MPR nodes are connected to all nodes within a two-hop radius (see Figure 2). The information about MPR is updated whenever changes are detected in nodes that are one or two hops away from the node.

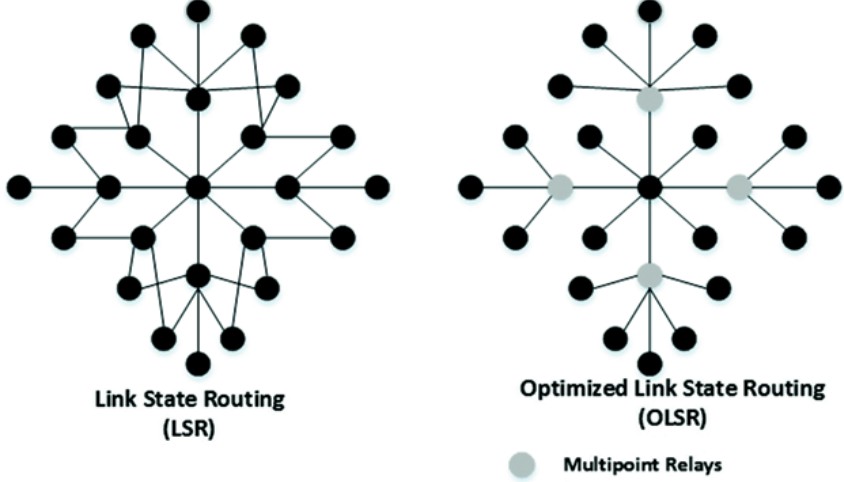

**Figure 2.** Route construction using MPR in the OLSR protocol.

Each node in the network maintains its routing table, which is formed based on the network topology information. It is distributed throughout the network via topology control (TC) route selection service packets. Only MPR nodes participate in forward-

ing TC packets, while other nodes receive and process such packets but do not forward them further.

For each MPR node, a list of neighboring nodes that have selected it as an MPR, called the MPR Selectors (MPRS) list, is created. Information about MPRS is transmitted in special Hello packets that are transmitted only between two neighboring nodes. In the network (in TC packets), only information about the state of connections between an MPR and its MPRS is transmitted. This mechanism significantly reduces the amount of control packet transmissions compared to flooding. Additionally, OLSR control messages contain sequential numbers that increase in subsequent messages. Therefore, the recipient of a control message can easily determine which information is up-to-date. OLSR is designed as a fully distributed protocol and does not rely on any root nodes. Each node also periodically broadcasts control packets, ensuring protocol stability even in the event of partial message loss. The architecture and advantages of the OLSR base protocol have led to the development of numerous protocol modifications that support QoS in various forms.

OLSR is a table-driven protocol. It supports several active tables: for tracking symmetric neighbors with one transition, neighbors with two transitions, MPR, a topology information base, a repeating message set, a multiple interface union set, and more.

Based on input information from Hello packets and topology control packets, and information about its multiple network interfaces, each node creates sets of links, neighbor sets, two-hop neighbor sets, MPR sets, MPR selector sets, a topology information base, duplicate sets, and multiple interface join tables. Based on these tables, each node decides whether it will forward OLSR messages received by it to other neighbors. Based on the topology, it creates routes.

Figure 3 covers the OLSR routing protocol mechanism [26].

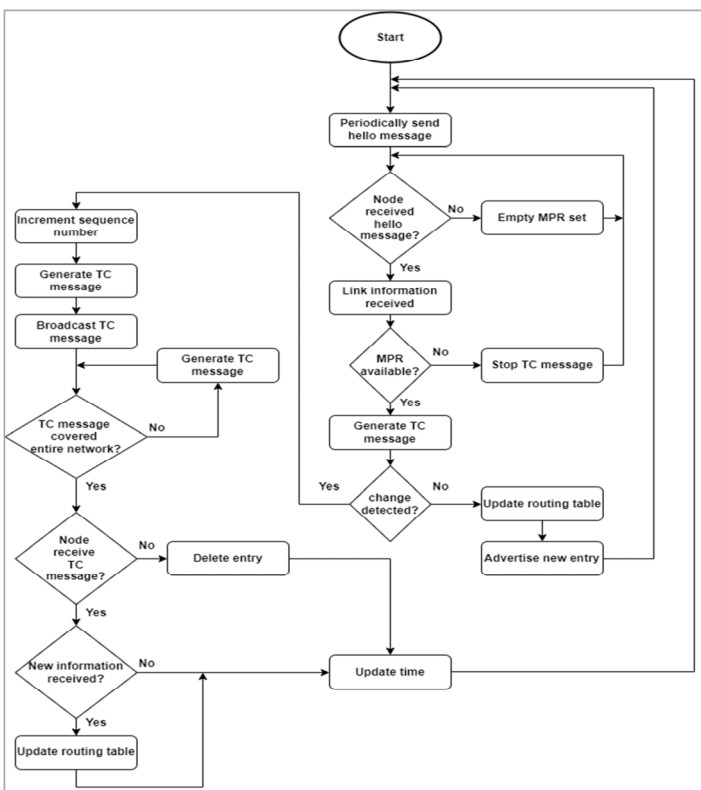

**Figure 3.** The OLSR routing protocol flowchart.

### 2.3. Resource Reservation Protocol (RSVP)

The quality control of packet transmission in an MANET is realized with a Resource Reservation Protocol (RSVP) [34]. The RSVP protocol allows applications to request the

desired level of the quality of service. It operates over the Internet Protocol (IP) and initiates resource reservations from the receiver's side.

The main characteristics of RSVP can be highlighted as follows:

1.    RSVP can work with various protocol stacks, but it is primarily designed for TCP/IP networks where data are transmitted as datagrams without establishing connections.
2.    The main purpose of the protocol is to provide QoS for selected data streams whose transmission has unacceptable latency or bandwidth requirements.
3.    RSVP runs on top of IPv4 or IPv6. RSVP is more of an Internet control protocol whose implementation runs in the background rather than in the data forwarding path.
4.    RSVP is a receiver-oriented signaling protocol. The receiver initiates and maintains the resource reservation.
5.    The protocol is used for both unicast and multicast transmissions.
6.    RSVP supports dynamic automatic adaptation to changes in the network.

RSVP requests resources for simplex flows, meaning it requests resources in only one direction. The transmission process begins when the data source (application) sends a PATH command to potential receivers. The command includes a stream identifier (which may contain the source address and other fields from the TCP/IP header) that allows the router to associate packets with a specific RSVP session. The PATH command also describes the expected information flow, such as specifying its volume. Each router processing the PATH command remembers the stream identifier and the incoming channel for that stream. This allows for the creation of tables of data flow paths and ensures that all routers are ready to allocate resources.

If a potential receiver, upon receiving the PATH command, wants to receive the specified data, it sends a RESV command. Using the same stream identifier, the RESV command follows the reverse path of the PATH command. It informs the routers on its path of the required QoS level. A router that receives RESV commands from multiple receivers combines them into a single command. If a router can provide the requested QoS level, it adds the reservation to the so-called flow table; otherwise, it denies resource reservation. Routers use flow tables to determine whether incoming datagrams belong to a particular RSVP session.

Since the resulting multicast topology may change over time, the RSVP design assumes that the RSVP state and traffic control state should be dynamically created and torn down in routers and hosts. To achieve this, RSVP establishes a "soft" state, meaning it sends periodic refresh messages to maintain the state along the reserved path(s). In the absence of refresh messages, the state is automatically deactivated and removed.

## 3. Results

### 3.1. Network Parameters Determined before Starting Work

The receiving and transmitting devices of the network nodes were smartphones or portable devices with the following characteristics.

### 3.1.1. Physical Layer Parameters

The physical layer parameters were chosen based on the prevalence of devices with such characteristics and their relatively low cost:

Transmitter power: 14 dBm; receiver sensitivity: −80 dBm

Data transmission rates for modulation and coding scheme (MCS index) number 7, the number of spatial streams (SS = 1), and the Guard Interval between symbols (GI) equal to 400 ns in accordance with the 802.11n and 802.11ac standards are shown in Table 2 [12].

**Table 2.** Transmission rates for the selected modulation and coding scheme.

| MCS index | | | 7 |
|---|---|---|---|
| Spatial streams, SS | | | 1 |
| Modulation scheme | | | 64-QAM |
| Coding | | | 5/6 |
| Data transmission rates (Mbit/s) | channel bandwidth, 20 MHz | GI = 400 ns | 72.2 |
| | channel bandwidth, 40 MHz | GI = 400 ns | 150 |
| | channel bandwidth, 80 MHz | GI = 400 ns | 325 |
| | channel bandwidth, 160 MHz | GI = 400 ns | 650 |

The main difference when using the 802.11ac standard with the same modulation and coding scheme is that it provides the capability to expand the channel by four and eight times (up to 80 and 160 MHz, respectively).

3.1.2. Network and Transport Layer Parameters

Routing is performed using the OLSR protocol. Quality of service control is achieved through the RSVP protocol. The quality of service for transmitted packets is determined with ITU-T recommendation Y.1541 [35]. Table 3 presents the acceptable IP packet delivery delays.

**Table 3.** Norms for the characteristics of IP networks with distribution by quality of service classes.

| Network Specifications | Quality of Service Classes | | | | | |
|---|---|---|---|---|---|---|
| | **0** | **1** | **2** | **3** | **4** | **5** |
| IP Packet Transfer Delay, IPTD | 100 ms | 400 ms | 100 ms | 400 ms | 1 s | U |

Note: U stands for undefined.

*3.2. Evaluation Metrics and Performance Analysis in MANETs*

We will use the following evaluation metrics to analyze the achievable performance of the MANET:

- Route acquisition time
- Routing overhead
- End-to-end delay
- Losses on the radio signal propagation path between nodes
- Network scalability

3.2.1. Route Acquisition Time

This characteristic is based on measuring the time required to establish the routing table on each network node and to determine the necessary bandwidth for a specific connection based on the class of transmitted traffic. To evaluate the overall route acquisition time $T_m$, we can use the following formula:

$$T_m = T_H + T_{TC} + T_R \qquad (1)$$

where

$T_H$—time required to receive Hello packets;
$T_{TC}$—time required to receive all TC packets;

$T_R$—time required to establish a channel with the necessary bandwidth between the end receiver node and the transmitting node.

$$T_H = 2 * X_H / R \tag{2}$$

where

$X_H = 86$ bytes—size of Hello packet (information about one-hop neighbors, transmitted to neighbors with one-hop link);
$R$—speed between the transmitting and receiving node.

$$T_{TC} = 2 * (N - 1)^2 * X_{TC} / R, \tag{3}$$

where

$N$—number of nodes in the network;
$X_{TC} = 74$ bytes—size of TC packet (information only about one-hop neighbors, transmitted only to MPR one-hop neighbors).

$$T_R = 2 * (n - 1) * X_R / R, \tag{4}$$

where

$n$—number of nodes from the transmitting end node to the receiving end node;
$X_R = 50 + 16 * (n - 1)$—size of RSVP packet.

It should be noted that the delay calculation formulas were designed to estimate the maximum possible route acquisition time in the network. Figure 4 illustrates the graph depicting the relationship between route acquisition time and the number of nodes in the network. The upper curve corresponds to the network operating in the 802.11ac standard, while the lower curve corresponds to the network operating in the 802.11n standard.

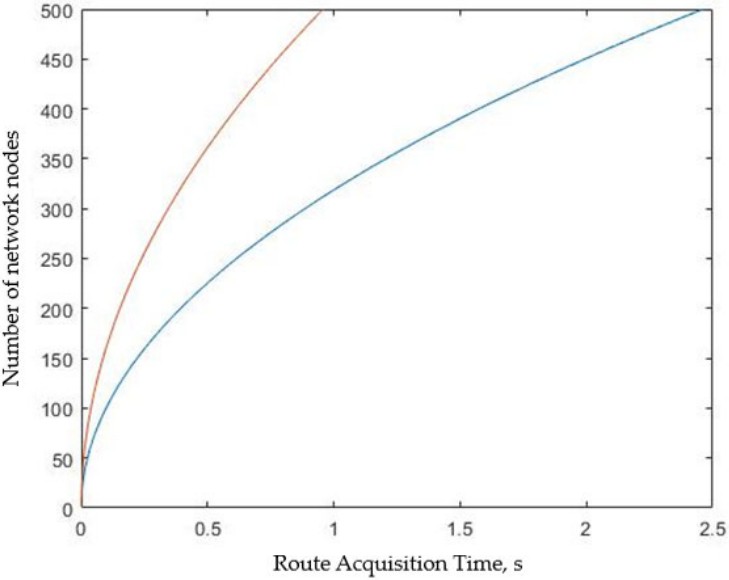

**Figure 4.** Dependency of route acquisition time on the number of nodes.

As observed from the graph, as the number of nodes in the network increases, more time is required to establish the route. For convenience, Table 4 presents characteristic times for specific numbers of nodes.

**Table 4.** Dependence of the Route Acquisition Time on the number of network nodes of a fixed number of nodes.

| Number of Nodes | Route Acquisition Time $T_m$, s |
| :---: | :---: |
| 50 | 0.01–0.02 |
| 75 | 0.02–0.05 |
| 100 | 0.04–0.1 |
| 200 | 0.15–0.39 |
| 350 | 0.5–1.24 |

During the calculation of $T_m$, a speed value reflecting the maximum achievable speed between nodes in different standards was used. Depending on the nature of the traffic, network purpose, topology, and other factors, it may be more appropriate to utilize only a portion of the available bandwidth between nodes or allocate the bandwidth for signaling traffic at specific times. At the same time, the route acquisition time will also increase, which, in turn, will affect how much time it takes to transfer data over the network, the fault tolerance of the network, and its capabilities.

3.2.2. Routing Overhead

Routing overhead is calculated based on the proportion of signaling and useful traffic. Signaling traffic includes a routing protocol (OLSR) and quality of service (RSVP) traffic. To assess efficiency, the ratio of signaling traffic to the maximum amount of useful traffic per minute is considered, depending on the number of network nodes:

$$E = (q * X_H + m * X_{TC} + k * X_R) / (60 * R - (q * X_H + m * X_{TC} + k * X_R)) \qquad (5)$$

The calculations involve the following values for the transmitted information in the network:

Standard WPA2 (CCMP) header: 50 bytes;
OLSR header: 16 bytes;
Hello packet length: ($8 + n * 4$, where $n$ is the number of neighbors) bytes;
TC message packet length: ($4 + m * 4$, where $m$ is the number of MPR neighbor addresses) bytes;
IP header: 20 bytes and UDP header: 4 bytes;
Payload size (maximum and minimum, depending on packet length): 576 bytes;
OLSR signaling information is periodically sent by nodes, with default broadcast intervals: the Hello packet interval is 2 s, and the TC packet interval is 5 s.

Figure 5 in a graph presents the results of calculations for a single active network node. The upper curve corresponds to the practical speed value when the network operates in the 802.11ac standard, while the lower curve corresponds to the network operating in the 802.11n standard.

From the graph, it can be observed that for up to 500 nodes in the 802.11n standard and up to 1000 nodes in the 802.11ac standard, the proportion of signaling information does not exceed 5% of the useful traffic.

It is worth noting that when calculating the routing overhead for the entire network, it is necessary to consider that almost every node also passes through itself the signaling traffic of other nodes. For a specific network topology, it is possible to perform calculations taking into account the characteristics of routing protocols, communication channels between receiver and transmitter nodes, and other factors that influence the amount of transmitted signaling traffic.

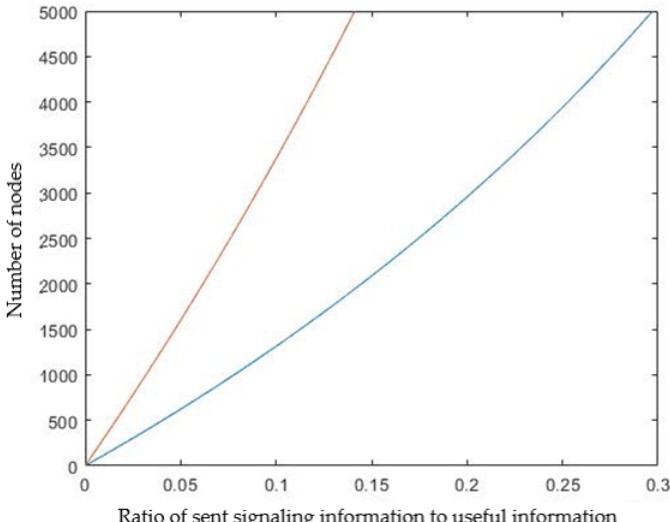

**Figure 5.** The dependence of the routing overhead on the number of network nodes.

### 3.2.3. End-to-End Delay in Network Transmission

End-to-end delay (ETED), also known as packet delay (PD) in network transmission, is influenced by various factors such as the number of nodes in the network, network topology, the number of active transmitting nodes, the number of receiving nodes, etc. This section analyzes the impact of these conditions on the delivery time of an IP packet from an end transmitting node to an end receiving node.

To calculate the end-to-end delay, the following network scenarios were considered:

1.  Each node in the network generates its own packets and participates in packet transmission. It must transmit to some node a packet within the transmission route from any transmitter terminal node to any receiver terminal node. The maximum delay dependence on the number of nodes in this case is given using the expression

$$T = 2 * \left(\frac{X}{R}\right) * \left(\frac{n}{2}\right)^2 \tag{6}$$

*T*—time delay for delivering an IP packet;
*X*—packet size including headers;
*R*—speed between nodes;
*n*—number of nodes participating in packet transmission.

Delay calculations are performed for different quality of service values, such as 0.1 s, 0.4 s, and 1 s. Table 5 presents the dependencies of PD on the number of nodes for various transmission speeds (channel bandwidths).

**Table 5.** The dependencies of delay on the number of nodes for various transmission speeds (channel bandwidths).

| Packet Delay, s | Number of Nodes for Channels with Bandwidth 20 MHz | Number of Nodes for Channels with Bandwidth 40 MHz | Number of Nodes for Channels with Bandwidth 80 MHz | Number of Nodes for Channels with Bandwidth 160 MHz |
|---|---|---|---|---|
| 0.1 | 36 | 47 | 76 | 106 |
| 0.4 | 74 | 90 | 152 | 214 |
| 1 | 112 | 150 | 242 | 338 |

As the channel bandwidth increases, it allows for transmission through a larger number of nodes. Thus, to achieve different quality of service classes, only a limited number of nodes may be used.

2.  Each node in the network generates traffic and transmits packets to a gateway end node in other networks.

$$T = 2 * \left( \frac{X}{R} \right) * \left( \left( n^2 - n \right) / 2 \right) \tag{7}$$

The results of calculations for this scenario are presented in Table 6.

**Table 6.** The dependencies of delay on the number of nodes for various transmission speeds.

| Packet Delay, s | Number of Nodes for Channels with Bandwidth 20 MHz | Number of Nodes for Channels with Bandwidth 40 MHz | Number of Nodes for Channels with Bandwidth 80 MHz | Number of Nodes for Channels with Bandwidth 160 MHz |
|---|---|---|---|---|
| 0.1 | 24 | 35 | 52 | 74 |
| 0.4 | 50 | 72 | 108 | 148 |
| 1 | 80 | 114 | 170 | 234 |

Comparing Tables 5 and 6, it can be observed that when the traffic propagation scenario changes, the maximum number of nodes in the network with acceptable packet delivery delay significantly decreases (by approximately 1.3 times).

3.  Different numbers of participant nodes generate traffic for the gateway node.

$$T = 2 * \left( \frac{X}{R} \right) * \left( k * \frac{\left( n^2 - n \right)}{2} - (1 - k) * n \right), \tag{8}$$

where

$k$—"Active Nodes Percentage"; represents the proportion of active nodes that generate their own packets.
$(1 - k)$—"Passive Nodes Percentage"; refers to the number of passive nodes that simply relay received packets along their route.

For the calculations, four ratios of active and passive nodes are considered: 1.0, 0.75, 0.5, and 0.25. The results are summarized in Tables 7 and 8 for channel bandwidths between nodes of 40 MHz and 80 MHz, respectively.

**Table 7.** The packet delay depending on the ratio of active and passive nodes.

| Packet Delay, s | Number of Nodes for Channels with Bandwidth 40 MHz 100% of Nodes Are Transmitters | Number of Nodes for Channels with Bandwidth 40 MHz 75% of Nodes Are Transmitters | Number of Nodes for Channels with Bandwidth 40 MHz 50% of Nodes Are Transmitters | Number of Nodes for Channels with Bandwidth 40 MHz 25% of Nodes Are Transmitters |
|---|---|---|---|---|
| 0.1 | 35 | 40 | 48 | 66 |
| 0.4 | 72 | 78 | 94 | 130 |
| 1 | 106 | 122 | 150 | 212 |

**Table 8.** The packet delay depending on the ratio of active and passive nodes.

| Packet Delay, s | Number of Nodes for Channels with Bandwidth 80 MHz 100% of Nodes Are Transmitters | Number of Nodes for Channels with Bandwidth 80 MHz 75% of Nodes Are Transmitters | Number of Nodes for Channels with Bandwidth 80 MHz 50% of Nodes Are Transmitters | Number of Nodes for Channels with Bandwidth 80 MHz 25% of Nodes Are Transmitters |
|---|---|---|---|---|
| 0.1 | 52 | 62 | 76 | 100 |
| 0.4 | 108 | 124 | 152 | 212 |
| 1 | 170 | 196 | 244 | 340 |

According to the data provided in the above tables, the number of relay nodes has a direct proportional effect on the number of nodes in the data transmission route. It is worth noting that the relay nodes should not transmit their own packets until the packet required for relay is delivered to the next two-hop neighbor along the route.

4. Multiple branches with the same number of nodes generate traffic for the gateway node:

$$T_1 = 2 * \left(\frac{X}{R}\right) * \left(\frac{n}{2}\right)^2, \tag{9}$$

$$T_2 = 2 * \left(\frac{X}{R}\right) * \left(\frac{\left(\left(\frac{n}{2}\right)^2 - \frac{n}{2}\right)}{2} + \left(\frac{n}{2} - 1\right)\right), \tag{10}$$

$$T_3 = 2 * \left(\frac{X}{R}\right) * \left(\frac{\left(\left(\frac{n}{3}\right)^2 - \frac{n}{3}\right)}{2} + 2 * \left(\frac{n}{3}, -, 1\right)\right), \tag{11}$$

where

$T_1$—ETTD on packet delivery for a single network branch;
$T_2$—ETTD for packet delivery using two branches in the network;
$T_3$—ETTD for packet delivery using three branches in the network.

Tables 9 and 10 demonstrate the impact of data transmission path branching on the packet delivery delay for the scenario where all nodes send packets to a single gateway node.

**Table 9.** The effect of branching of the data transmission path on the delay in the delivery of packets in cases where all nodes send packets to the same gateway node.

| Packet Delay, s | Number of Nodes with 40 MHz Channel Bandwidth, One Branch | Number of Nodes with 40 MHz Channel Bandwidth, Two Branches | Number of Nodes with 40 MHz Channel Bandwidth, Three Branches |
|---|---|---|---|
| 0.1 | 35 | 66 | 96 |
| 0.4 | 72 | 132 | 196 |
| 1 | 106 | 210 | 314 |

**Table 10.** The effect of branching of the data transmission path on the delay in the delivery of packets in cases where all nodes send packets to the same gateway node.

| Packet Delay, s | Number of Nodes with 80 MHz Channel Bandwidth, One Branch | Number of Nodes with 80 MHz Channel Bandwidth, Two Branches | Number of Nodes with 80 MHz Channel Bandwidth, Three Branches |
|---|---|---|---|
| 0.1 | 52 | 110 | 158 |
| 0.4 | 108 | 212 | 315 |
| 1 | 170 | 336 | 653 |

The calculations indicate that the branching of the network significantly affects the transmission of packets from the source node to the gateway node, allowing for a greater number of connected nodes as the number of branches increases, while providing the required delay for different classes of traffic.

5.  The final scenario, in which half of the nodes transmit packets for the other half of the network nodes, i.e., each node on the path generates traffic for only one other node. Several transmission branches are symmetrical:

$$T_1 = 2 * \left(\frac{X}{R}\right) * \left(\left(n^2 - n\right)/2\right), \tag{12}$$

$$T_2 = 2 * \left(\frac{X}{R}\right) * \left(\frac{n}{2}\right)^2, \tag{13}$$

$$T_3 = 2 * \left(\frac{X}{R}\right) * \left(\left(\frac{\left(n - \frac{n}{3}\right)}{2}\right)^2 + \left(n - \frac{n}{3} - 1\right)\right), \tag{14}$$

where

$T_1$—ETTD for a single branch in the network.
$T_2$—ETTD for two branches in the network.
$T_3$—ETTD for three branches in the network.

Tables 11 and 12 demonstrate the impact of data transmission path branching on the packet delivery delay for scenarios where nodes exchange data pairwise.

**Table 11.** The dependence of the influence of branching of the data transmission path on the delay in the delivery of packets for 40 MHz channels.

| Packet Delay, s | Number of Nodes with 40 MHz Channel Bandwidth, One Branch | Number of Nodes with 40 MHz Channel Bandwidth, Two Branches | Number of Nodes with 40 MHz Channel Bandwidth, Three Branches |
|---|---|---|---|
| 0.1 | 46 | 68 | 86 |
| 0.4 | 96 | 140 | 176 |
| 1 | 152 | 220 | 292 |

Comparing the last two packet transmission scenarios, it can be observed that the number of branches has a stronger influence on the maximum number of nodes in networks where nodes exchange packets among themselves. However, in the case of three branches, there is a reverse effect. This is due to the fact that in the latter case, the number of nodes that have three one-step neighbors is two instead of one as in the previous case.

**Table 12.** The dependence of the influence of branching of the data transmission path on the delay in the delivery of packets for 80 MHz channels.

| Packet Delay, s | Number of Nodes with 80 MHz Channel Bandwidth, One Branch | Number of Nodes with 80 MHz Channel Bandwidth, Two Branches | Number of Nodes with 80 MHz Channel Bandwidth, Three Branches |
|---|---|---|---|
| 0.1 | 76 | 110 | 140 |
| 0.4 | 150 | 224 | 294 |
| 1 | 240 | 360 | 484 |

It should be noted that the increase in the number of branches in a node is limited by the network parameters, such as having two or three neighboring one-hop nodes for each node. Therefore, in this calculation, it is considered that only three branches will lead to the "central" node. In a different data transmission path topology, it is important to consider that if each node has three one-hop neighbors, it will result in the end nodes being in close proximity to each other, which limits the size of the network.

From this, it can be concluded that an effective data transmission path topology for covering a larger area would be a network where some nodes have two one-hop neighbors, while others have three one-hop neighbors.

3.2.4. Losses on the Radio Signal Propagation Path between Nodes

When designing the network, transmitter radio visibility calculations were performed in accordance with ITU-R Recommendation P.1411-11 [36] and ITU-R Recommendation P.1238-10 [37]. We will consider scenarios where the network is located outdoors in an urban area and inside residential buildings.

First, let us analyze the main transmission loss in radio wave propagation within street canyons, where the receiving and transmitting stations are located below roof level, regardless of the height of their antennas. In this case, the median main transmission loss is determined using the following formula [36]:

$$L(d, f) = 10 * \alpha * \log_{10} d + \beta + 10 * \gamma * \log_{10} f, \text{ dB} \tag{15}$$

with a standard deviation ($\sigma$) (in dB).

$d$—distance between the receiving and transmitting stations (m);
$f$—operating frequency (GHz);
$\propto$, $\beta$, $\gamma$—coefficients depending on the type of route.

There are three possible cases when designing the network. The first case involves stations located within the line of sight (LOS). For such paths and types of environments—urban high-rise buildings, urban areas with low-rise buildings, and suburban areas—the following coefficient values are recommended:

$$\propto = 2.12 \; \beta = 29.2 \; \gamma = 2.11 \text{ и } \sigma = 5.06 \text{ dB}$$

The second case involves non-line-of-sight (NLOS) propagation in urban conditions. In the case of network deployment in residential areas with single-story and two-story residential buildings, the loss formula incorporates the following coefficients:

$$\propto = 3.01 \; \beta = 18.8 \; \gamma = 2.07 \text{ и } \sigma = 3.07 \text{ dB}$$

When deploying the network in urban areas with high-rise buildings, the following are incorporated:

$$\propto = 4.00 \; \beta = 10.2 \; \gamma = 2.36 \text{ и } \sigma = 7.06 \text{ dB}$$

When nodes in the network are located inside a building, it is advisable to use a generalized model of the main transmission losses inside the premises in accordance with ITU-R Recommendation P.1238-10 [37]:

$$L(d, f) = 20 * \log_{10} f + N * \log_{10} d + L_f(n) - 28 \text{ dB} \tag{16}$$

$f$—channel central frequency (MHz);
$d$—distance between neighboring nodes (m), where d > 1 m;
$N$—distance loss factor;
$L_f(n)$—loss due to signal penetration through walls;
$n$—number of floors between nodes, where $n = 0$ corresponds to adjacent floors and $L_f(n) = 0$ dB. For a frequency of 2.4 GHz, $L_f(n) = 10$ dB (for a single concrete wall) and $N = 28$; for a frequency of 5 GHz, $L_f(n) = 13$ dB and $N = 30$.

Tables 13–16 present the signal loss (attenuation) at frequencies of 2.4 GHz and 5 GHz for different scenarios.

**Table 13.** Signal attenuation (loss) for line of sight paths.

| L, dB | d(m), at f = 2.4 GHz | d(m), at f = 5 GHz |
|---|---|---|
| −90...−80 | 104...310 | 48...149 |
| −80...−70 | 35...104 | 17...48 |
| −70...−60 | 12...35 | 6...17 |
| −60...−50 | 4...12 | 1,8...6 |

**Table 14.** Signal attenuation (loss) for NLoS paths in the residential zone with single-story and two-story residential buildings.

| L, dB | d(m), at f = 2.4 GHz | d(m), at f = 5 GHz |
|---|---|---|
| −90...−80 | 58...127 | 37...77 |
| −80...−70 | 28...58 | 17...37 |
| −70...−60 | 13...28 | 8...17 |
| −60...−50 | 6...13 | 4...8 |

**Table 15.** Signal attenuation (loss) for NLoS paths in urban areas with multi-story buildings.

| L, dB | d(m), at f = 2.4 GHz | d(m), at f = 5 GHz |
|---|---|---|
| −90..−80 | 33..59 | 21..39 |
| −80..−70 | 19..33 | 12..21 |
| −70..−60 | 11..19 | 7..12 |
| −60..−50 | 6..11 | 4..7 |

**Table 16.** Signal loss when placing network nodes indoors on adjacent floors, taking into account attenuation through a single concrete wall.

| L, dB | d(m), at f = 2400 MHz | d(m), at f = 5000 MHz |
|---|---|---|
| −90...−80 | 12...28 | 7...17 |
| −80...−70 | 5...12 | 3...7 |
| −70...−60 | 3...5 | 1...3 |
| −60...−50 | 1...3 | 0...1 |

Taking into account the specified transmitter power of 14 dBm and receiver sensitivity of −80 dBm for the nodes, we can determine the possible range of distances between nodes where the signal will have an acceptable attenuation to ensure the required data transmission rate. Within indoor environments (residential buildings), this distance would be either 25 or 15 m (depending on the central frequency). When nodes are located outside residential buildings, the possible distances between nodes are as follows: (1) for line of sight paths in urban areas, up to 300 or 150 m; (2) for NLoS paths in residential areas, 70 or 50 m; and (3) for NLoS paths in urban areas with high-rise buildings, 50 or 30 m.

## 4. Discussion

Let us discuss the issues related to network scalability. Network scalability in this context refers to the assessment of the potential distance and area over which the network can reliably and qualitatively transmit data of different service classes. Factors influencing network scalability include the allowable distance between nodes and the maximum distance between end transmitter and receiver nodes.

The scalability of the network is influenced by the factors analyzed above: route acquisition time, routing overhead, end-to-end delay, network topology and data transmission paths, and signal loss along the transmission path between nodes.

The calculated value of route acquisition time ranges from 100 ms to 1.5 s, depending on the number of nodes in the network (from 50 to 500, respectively). Considering the quality of service requirements and other characteristics, this duration can be considered satisfactory. However, this calculation assumes that nodes do not transmit other packets during the route establishment. Therefore, if there is channel congestion between nodes, the route acquisition time may increase.

The solution to this problem is possible with the help of certain network settings, in which not all the bandwidth of the channel is allocated for route finding. The size and frequency of sending routing packets can also be defined. This will reduce the impact of transmitted service packets on the channel bandwidth.

The routing overhead, expressed as the ratio of useful packets to control packets transmitted per minute by a single node, indicates that each node generates no more than 5% of control traffic out of 100% of the transmitted traffic, with the number of other nodes ranging from 500 to 1000, depending on the bandwidths of 40 MHz and 80 MHz. However, if the number of control packets to be sent to not only the nearest neighbors (as considered in the calculations) but also for topology control messages from other nodes needs to be taken into account, this ratio may increase for different network topologies.

The main method aimed at mitigating the impact of this factor is the MPR technology, which significantly reduces the number of control messages transmitted for network topology control. By selecting the most connected nodes to other nodes, conventionally speaking, a network graph tree is constructed in which all terminal nodes transmit only one topology control message each to the most connected nodes, which propagate the topology information among themselves and with the terminal nodes. The most favorable topology in terms of minimizing overhead transmission is to have a significant number of branches from central nodes. However, the central nodes themselves should not form an extended line.

Unlike the previous conclusion, it can be stated with certainty that the path of packet transmission from the terminal node transmitter to the terminal node receiver forms an extended line of multiple nodes with possible branching. The paper shows that depending on the class of the quality of service and the bandwidth of the connection between the nodes, the number of nodes on the path of data packet transmission can vary from 24 to 238 nodes, assuming a full load of the connections between the nodes. With the presence of relay nodes that do not generate traffic during packet transmission, the maximum number of nodes can be increased to 340. It should also be noted that in the scenario of packet transmission from all nodes to one gateway node, it is possible to increase the number of nodes on the packet transmission route by approximately one and a half to two times.

The network topology also significantly affects network scalability. Provided that the part of the nodes carrying traffic through the path from the transmitter end node to the source end node is in a branch from one of the nodes, it is possible to increase the number of nodes by a factor of two or three. However, it should be taken into account that the more branches there are in the network, the larger the area it covers, but the distance between the most distant nodes will decrease.

Losses also impose limitations on scalability. The maximum acceptable distance between nodes ranges from 10 to 56 m, depending on the connection throughput between nodes, node locations, and central frequency.

Thus, by knowing the node locations (outdoors in urban or rural areas or inside residential buildings), the allowable distance between nodes, and the number of nodes that ensure the desired quality of service for each scenario, it is possible to assess the possible distances and areas over which the network can reliably and qualitatively transmit data.

Analytical expressions for calculating the values of metrics characterizing the network operation are obtained in this paper. Calculations using these expressions are convenient to perform, for example, in MATCAD or MATLAB environments. In this case, it is possible to immediately build dependencies of performance indicators on the network parameters. Therefore, the computational cost of the study was minimal compared to modeling.

## 5. Conclusions and Future Work

The conducted analysis can be practically applied in the design of MANETs to assess network parameters capable of providing information transmission within a specified service class at a given distance in different network deployment locations.

The recommended maximum distances between nodes assume a high node density ranging from 15 m to 25 m indoors (depending on the frequency used) and 50 m to 70 m outdoors.

By utilizing the 802.11ac standard, network speed can be increased through channel expansion and reduced interference by utilizing the 5 GHz central frequency. Data transmission synchronization among different nodes can also improve throughput and reduce network load. Network scalability heavily depends on the quality of service requirements for routed packets, the number of senders, the throughput between participating nodes, network topology, and network operation scenarios. The maximum possible distance between the source and receiver with a packet delivery delay of 100 ms is approximately 5 km, 10 km for a delay of 400 ms, and 16 km for a delay of 1 s.

The results of calculations of MANET parameters using the expressions obtained in the paper are confirmed with the modeling results given in [31,38,39].

In further work, it is intended to focus on the creation, on the basis of the expressions obtained in the paper, of the methodology of the reasonable preliminary selection of MANET parameters for modeling or experimental studies of networks with routing OLSR, designed to solve specific problems in different environments.

**Funding:** This research received no external funding.

**Data Availability Statement:** The data presented in this study are available on request from the corresponding author.

**Conflicts of Interest:** The author declares no conflict of interest.

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
