# Peer review of "Estimation of the Achievable Performance of Mobile Ad Hoc Networks with Optimal Link State Routing"

_inventions, doi:10.3390/inventions8050108_

Round 1

Reviewer 1 Report

The author provides a detailed view of the performance of the MANET network organization by analyzing the contribution of each elementary networking action. Useful insight is given of the wireless packet communication system performance at the absence of central nodes. On the other hand, the approach lacks experimental verification data, while no information is provided on the special hardware components potentially being used to support this study.

Title should be shortened and become more compact and accurate.

Section 1, has quite obsolete and few references about the topic being discussed.

In general the references should be enriched with more material, either “fresh”/state-of-the-art or more complete from the theoretical and the applied perspective. In this regard, the author, may find beneficial the material described in:

1)    Eltahlawy, A.M.; Aslan, H.K.; Abdallah, E.G.; Elsayed, M.S.; Jurcut, A.D.; Azer, M.A. A Survey on Parameters Affecting MANET Performance. Electronics 2023, 12, 1956. https://doi.org/10.3390/electronics12091956

2)  Pinitkan, S.; Wisitpongphan, N. Abnormal Activity Detection and Notification Platform for Real-Time Ad Hoc Network. Int. J. Onl. Eng. 2020, 16, pp. 45-63. https://doi.org/10.3991/ijoe.v16i15.16065

 3)      Ioannis Manolopoulos, Kimon Kontovasilis, Ioannis Stavrakakis, Stelios C.A. Thomopoulos, Methodologies for calculating decision-related event occurrence times, with applications to effective routing in diverse MANET environments, Ad Hoc Networks, Volume 99, 2020, 102068, ISSN 1570-8705, https://doi.org/10.1016/j.adhoc.2019.102068.

4)  Manolopoulos, I.; Loukatos, D.; Kontovasilis, K. A Versatile MANET Experimentation Platform and Its Evaluation through Experiments on the Performance of Routing Protocols under Diverse Conditions. Future Internet 2022, 14, 154. https://doi.org/10.3390/fi14050154

5)  Mangasuli, S. and Kaluti, M., 2023. Efficient Multimedia Content Transmission Model for Disaster Management using Delay Tolerant Mobile Adhoc Networks. International Journal of Advanced Computer Science and Applications, 14(1). http://dx.doi.org/10.14569/IJACSA.2023.0140152

6)      N.S. Saba Farheen, Anuj Jain, Improved routing in MANET with optimized multi path routing fine tuned with hybrid modeling, Journal of King Saud University - Computer and Information Sciences,

Volume 34, Issue 6, Part A, 2022, Pages 2443-2450, ISSN 1319-1578, https://doi.org/10.1016/j.jksuci.2020.01.001.

The performance evaluation/results section, should include approaches involving simulator or emulator or real testbed environment description and/or tests.  

All references should be put at the same format.

English grammar/typos and style issues should be fixed.

Reviewer 2 Report

Although the subject of the article is quite interesting, the contribution of this work to science is not adequately justified. To this end, the introduction of this article should be enriched with more up to date references, instead of the outdated ones cited in the manuscript, so as to support the originality of the proposed solution.

The methodology (Section 2) which is followed in this work needs to be further analysed so as to be in use for other researchers. Moreover some Figures need to be further processed so as to improve their resolution (i.e. Figure 1).

The results are adequately presented and thoroughly interpreted in perspective of the working hypotheses so as other researchers to be permitted to reproduce certain aspects of them. In the Discussion section the findings of the research as well as their implications are discussed in a broad context, yet it could take some improvements.

Finally, in the conclusions section it would for the readership to be informed about some directions for future research.

The paper is written in appropriate English language according to the standards of the Journal, however some minor spell-checking might be of use.

Reviewer 3 Report

The authors focus their study on self organizing wireless mobile networks by utilizing the optimal link state routing with multi point relay optimization and focusing on the quality control through the resource reservation protocol. The authors have provided detailed expressions in order to calculate the achievable network characteristics by including and focusing on the router position time, the network efficiency, the packet transmission delay, and the signal propagation losses. The manuscript is overall well written and easy to follow and the authors have well thought out their main contributions. The provided theoretical analysis is concrete, complete, and correct and the authors have provided all the intermediate steps in order to enable the average reader to easily follow it. Furthermore, the provided numerical results are rich in order to demonstrate the pure  operation and the performance of the proposed framework. The authors are encouraged to consider the following suggestions provided by the reviewer in order to improve the scientific depth of their manuscript, as well as they need to address the following minor comments in order to further improve the quality of presentation of their manuscript. Initially, in Section 1, the authors need to elaborate on the current state-of-the-art in the field, e.g., Grieco, Luigi Alfredo, et al., eds. Ad-Hoc, Mobile, and Wireless Networks: 19th International Conference on Ad-Hoc Networks and Wireless, ADHOC-NOW 2020, Proceedings. Vol. 12338. Springer Nature, 2020 where similar research approaches have been introduced mainly focusing on the same target study. In section three, the authors need to include a table summarizing the main notation that has been used in the paper, which currently is quite excessive. In Section 4, the authors need to include an additional subsection discussing the implementation cost and the computational complexity of the proposed study. Furthermore, in Section 4, the authors need to provide some comparative evaluation results to similar approaches that have been introduced in the state-of-the-art in order to quantify the drawbacks and benefits of the proposed model. Finally, the overall manuscript needs to be checked for typos, syntax, and grammar errors in order to improve the quality of its presentation.

The overall manuscript needs to be checked for typos, syntax, and grammar errors in order to improve the quality of its presentation.

Round 2

Reviewer 1 Report

The author has incorporated the proposed suggestions in an sufficient manner. Reviewer feels covered.

PS: It is preferable each main word in the title to start with capital letter. 

Reviewer 2 Report

As it appears from studying the revised version of the article, the authors took intoconsideration the reviewer's comments and suggestions, presenting a substantially improved manuscript compared to the original one.According to the reviewer's opinion no additional revisions are required.

Reviewer 3 Report

The authors have addressed the reviewers' comments.